# Learning Sample Difficulty from Pre-trained Models for Reliable Prediction

**Peng Cui[1 5], Dan Zhang[2 3], Zhijie Deng[4]\*, Yinpeng Dong[1 5], Jun Zhu[1 5]\***

[1]Dept. of Comp. Sci. & Tech., Institute for AI, BNRist Center, THBI Lab,
Tsinghua-Bosch Joint ML Center, Tsinghua University, Beijing, 100084 China
[2]Bosch Center for Artificial Intelligence  [3] University of Tübingen
[4]Qing Yuan Research Institute, Shanghai Jiao Tong University    [5]RealAI
xpeng.cui@gmail.com, dan.zhang2@de.bosch.com, zhijied@sjtu.edu.cn
dongyinpeng@tsinghua.edu.cn, dcszj@tsinghua.edu.cn

## Abstract

Large-scale pre-trained models have achieved remarkable success in many applications, but how to leverage them to improve the prediction reliability of downstream models is undesirably under-explored. Moreover, modern neural networks have been found to be poorly calibrated and make overconfident predictions regardless of inherent sample difficulty and data uncertainty. To address this issue, we propose to utilize large-scale pre-trained models to guide downstream model training with sample difficulty-aware entropy regularization. Pre-trained models that have been exposed to large-scale datasets and do not overfit the downstream training classes enable us to measure each training sample's difficulty via feature-space Gaussian modeling and relative Mahalanobis distance computation. Importantly, by adaptively penalizing overconfident prediction based on the sample difficulty, we simultaneously improve accuracy and uncertainty calibration across challenging benchmarks (e.g., $+0.55\%$ ACC and $-3.7\%$ ECE on ImageNet1k using ResNet34), consistently surpassing competitive baselines for reliable prediction. The improved uncertainty estimate further improves selective classification (abstaining from erroneous predictions) and out-of-distribution detection.

## 1 Introduction

Large-scale pre-training has witnessed pragmatic success in diverse scenarios and pre-trained models are becoming increasingly accessible [9, 4, 3, 14, 44]. The community has reached a consensus that by exploiting big data, pre-trained models can learn to encode rich data semantics that is promised to be generally beneficial for a broad spectrum of applications, e.g., warming up the learning on downstream tasks with limited data [45], improving domain generalization [24] or model robustness [19], and enabling zero-shot transfer [54, 43].

This paper investigates on a new application – leveraging pre-trained models to improve the calibration and the quality of uncertainty quantification of downstream models, both of which are crucial for reliable model deployment in the wild [20, 13, 28, 32]. Model training on the task dataset often encounters ambiguous or even distorted samples. These samples are difficult to learn from – directly enforcing the model to fit them (i.e., matching the label with 100% confidence) may cause undesirable memorization and overconfidence. This issue comes from loss formulation and data annotation, thus using pre-trained models for finetuning or knowledge distillation with the cross-entropy loss will not solve the problem. That said, it is necessary to subtly adapt the training objective for each sample according to its sample difficulty for better generalization and uncertainty quantification. We exploit

---

*Corresponding authors.

37th Conference on Neural Information Processing Systems (NeurIPS 2023).

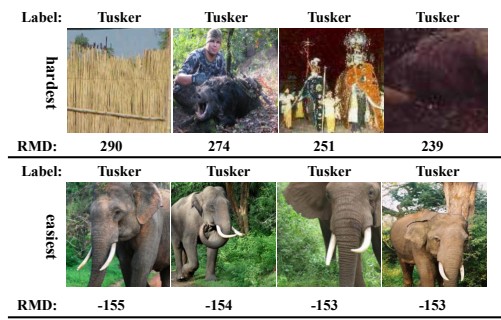

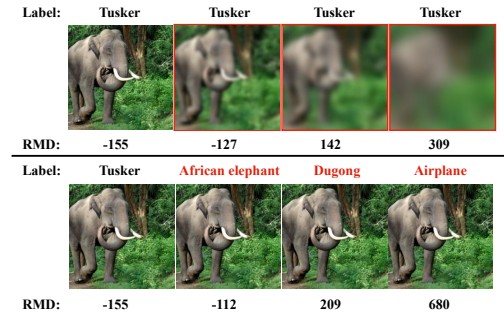

(a) Training samples from ImageNet class Tusker and ranked as hardest/easiest by CLIP-VIT-B.

(b) The sample difficulty score (RMD) increases proportionally with the corruption and label noise severity.

Figure 1: Visualization of training samples with low (high) sample difficulty that is scored by CLIP-ViT-B [44] using the derived relative Mahalanobis distance (RMD).

pre-trained models to measure the difficulty of each sample in the downstream training set and use this extra data annotation to modify the training loss. Although sample difficulty has been shown to be beneficial to improve training efficiency [33] and neural scaling laws [49], its effectiveness for ameliorating model reliability is largely under-explored.

Pre-trained models aid in scoring sample difficulty by shifting the problem from the raw data space to a task- and model-agnostic feature space, where simple distance measures suffice to represent similarities. Besides, large-scale multi-modal datasets and self-supervised learning principles enable the pre-trained models (see those in Table 1) to generate features that sufficiently preserve high-level concepts behind the data and avoid overfitting to specific data or classes. In light of this, we perform sample difficulty estimation in the feature space of pre-trained models and cast it as a density estimation problem since samples with typical discriminative features are easier to learn and typical features shall reappear. We advocate using Gaussian models to represent the training data distribution in the feature space of pre-trained models and derive the relative Mahalanobis distance (RMD) as a sample difficulty score. As shown in Fig. 1, there is a high-level agreement on sample difficulty between RMD and human cognition.

Equipped with the knowledge of sample difficulty, we further propose to use it for regularizing the prediction confidence. The standard cross entropy loss with a single ground truth label would encourage the model to predict all instances in Fig. 1-a) as the class Tusker with $100\%$ confidence. However, such high confidence is definitely unjustified for the hard samples. Thus, we modify the training loss by adding an entropic regularizer with an instance-wise adaptive weighting in proportion to the sample difficulty. Profiting from the high-quality sample difficulty measure and the effective entropic regularizer, we develop a strong, scalable, and easy-to-implement approach to improving both the predictive accuracy and calibration of the model.

Our method successfully improves the model's performance and consistently outperforms competitive baselines on various image classification benchmarks, ranging from the i.i.d. setting to corruption robustness, selective classification, and out-of-distribution detection. Importantly, unlike previous works that compromise accuracy [22, 31] or suffer from expensive computational overhead [26], our method can improve predictive performance and uncertainty quantification concurrently in a computationally efficient way. The consistent gains across architectures demonstrate that our sample difficulty measure is a valuable characteristic of the dataset for training.

## 2   Related Works

**Uncertainty regularization.**   A surge of research has focused on uncertainty regularization to alleviate the overfitting and overconfidence of deep neural networks. $L_p$ Norm [22] and entropy regularization (ER) [42] explicitly enforce a small norm of the logits or a high predictive entropy. Label smoothing (LS) [36] interpolates the ground-truth one-hot label vector with an all-one vector, thus penalizing $100\%$ confident predictions. [35] showed that focal loss (FL) [30] is effectively an upper bound to ER. Correctness ranking loss (CRL) [34] regularizes the confidence based on the frequency of correct predictions during the training losses.

LS, $L_p$ Norm, FL and ER do not adjust prediction confidence based on the sample difficulty. The frequency of correct predictions was interpreted as a sample difficulty measure [50]. However, CRL only concerns pair-wise ranking in the same batch. Compared to the prior art, we weight the overconfidence penalty according to the sample difficulty, which proves to be more effective at jointly improving accuracy and uncertainty quantification. Importantly, the sample difficulty is derived from a data distribution perspective, and remains constant over training.

**Sample difficulty measurement.** Prior works measure the sample difficulty by only considering the task-specific data distribution and training model, e.g., [50, 21, 40, 2, 1, 41, 49, 11, 33]. As deep neural networks are prone to overfitting, they often require careful selection of training epochs, checkpoints, data splits and ensembling strategies. Leveraging the measurement for data pruning, [49] showed existing work still underperforms on large datasets, e.g., ImageNet1k. The top solution used a self-supervised model and scored the sample difficulty via K-means clustering.

We explore pre-trained models that have been exposed to large-scale datasets and do not overfit the downstream training set. Their witness of diverse data serves as informative prior knowledge for ranking downstream training samples. Furthermore, the attained sample difficulty can be readily used for different downstream tasks and models. In this work, we focus on improving prediction reliability, which is an under-explored application of large-scale pre-trained models. The derived relative Mahalanobis distance (RMD) outperforms Mahalanobis distance (MD) and K-means clustering for sample difficulty quantification.

## 3 Sample Difficulty Quantification

Due to the ubiquity of data uncertainty in the dataset collected from the open world [23, 6], the sample difficulty quantification (i.e., characterizing the hardness and noisiness of samples) is pivotal for reliable learning of the model. For measuring training sample difficulty, we propose to model the data distribution in the feature space of a pre-trained model and derive a relative Mahalanobis distance (RMD). A small RMD implies that 1) the sample is typical and carries class-discriminative features (close to the class-specific mean mode but far away from the class-agnostic mean mode), and 2) there exist many similar samples (high-density area) in the training set. Such a sample represents an easy case to learn, i.e., small RMD $\leftrightarrow$ low sample difficulty, see Fig. 1.

### 3.1 Large-scale pre-trained models

Large-scale image and image-text data have led to high-quality pre-trained vision models for downstream tasks. Instead of using them as the backbone network for downstream tasks, we propose a new use case, i.e., scoring the sample difficulty in the training set of the downstream task. There is no rigorously defined notion of sample difficulty. Intuitively, easy-to-learn samples shall reappear in the form of showing similar patterns. Repetitive patterns specific to each class are valuable cues for classification. Moreover, they contain neither confusing nor conflicting information. Single-label images containing multiple salient objects belonging to different classes or having wrong labels would be hard samples.

To quantify the difficulty of each sample, we propose to model the training data distribution in the feature space of large-scale pre-trained models. In the pixel space, data distribution modeling is prone to overfitting low-level features, e.g., an outlier sample with smoother local correlation can have a higher probability than an inlier sample [48]. On the other hand, pre-trained models are generally trained to ignore low-level information, e.g., semantic supervision from natural language or class labels. In the case of self-supervised learning, the proxy task and loss are also formulated to learn a holistic understanding of the input images beyond low-level image statistics, e.g., the masking strategy designed in MAE [14] prevents reconstruction via exploiting local correlation. Furthermore, as modern pre-trained models are trained on large-scale datasets with high sample diversities in many dimensions, they learn to preserve and structure richer semantic features of the training samples than models only exposed to the training set that is commonly used at a smaller scale. In the feature space of pre-trained models, we expect easy-to-learn samples will be closely crowded together. Hard-to-learn ones are far away from the population and even sparsely spread due to missing consistently repetitive patterns. From a data distribution perspective, the easy (hard)-to-learn samples should have high (low) probability values.

## 3.2 Measuring sample difficulty

While not limited to it, we choose the Gaussian distribution to model the training data distribution in the feature space of pre-trained models, and find it simple and effective. We further derive a relative Mahalanobis distance (RMD) from it as the sample difficulty score.

**Class-conditional and agnostic Gaussian distributions.** The downstream training set $\mathcal{D} = \{(x_i, y_i)\}_{i=1}^N$ is a collection of image-label pairs, with $x_i \in \mathbb{R}^d$ and $y_i \in \{1, 2, ..., K\}$ as the image and its label, respectively. We model the feature distribution of $\{x_i\}$ with and without conditioning on the class information. Let $G(\cdot)$ denote the penultimate layer output of the pre-trained model $G$. The class-conditional distribution is modelled by fitting a Gaussian model to the feature vectors $\{G(x_i)\}$ belonging to the same class $y_i = k$

$$P(G(x) \mid y = k) = \mathcal{N}\left(G(x) \mid \mu_k, \Sigma\right), \tag{1}$$

$$\mu_k = \frac{1}{N_k} \sum_{i:y_i=k} G(x_i), \quad \Sigma = \frac{1}{N} \sum_k \sum_{i:y_i=k} (G(x_i) - \mu_k)(G(x_i) - \mu_k)^\top, \tag{2}$$

where the mean vector $\mu_k$ is class-specific, the covariance matrix $\Sigma$ is averaged over all classes to avoid under-fitting following [46, 27], and $N_k$ is the number of training samples with the label $y_i = k$. The class-agnostic distribution is obtained by fitting to all feature vectors regardless of their classes

$$P(G(x)) = \mathcal{N}\left(G(x) \mid \mu_{\text{agn}}, \Sigma_{\text{agn}}\right), \tag{3}$$

$$\mu_{\text{agn}} = \frac{1}{N} \sum_i^N G(x_i), \quad \Sigma_{\text{agn}} = \frac{1}{N} \sum_i^N (G(x_i) - \mu_{\text{agn}})(G(x_i) - \mu_{\text{agn}})^\top. \tag{4}$$

**Relative Mahalanobis distance (RMD).** For scoring sample difficulty, we propose to use the difference between the Mahalanobis distances respectively induced by the class-specific and class-agnostic Gaussian distribution in (1) and (3), and boiling down to

$$\mathcal{RM}(x_i, y_i) = \mathcal{M}(x_i, y_i) - \mathcal{M}_{\text{agn}}(x_i), \tag{5}$$

$$\mathcal{M}(x_i, y_i) = -\left(G(x_i) - \mu_{y_i}\right)^\top \Sigma^{-1} \left(G(x_i) - \mu_{y_i}\right), \tag{6}$$

$$\mathcal{M}_{\text{agn}}(x_i) = -\left(G(x_i) - \mu_{\text{agn}}\right)^\top \Sigma_{\text{agn}}^{-1} \left(G(x_i) - \mu_{\text{agn}}\right). \tag{7}$$

Previous work [46] utilized RMD for OOD detection at test time, yet we explore it to rank each sample and measure the sample difficulty in the training set.

A small class-conditional MD $\mathcal{M}(x_i, y_i)$ indicates that the sample exhibits typical features of the sub-population (training samples from the same class). However, some features may not be unique to the sub-population, i.e., common features across classes, yielding a small class-agnostic MD $\mathcal{M}_{\text{agn}}(x_i)$. Since discriminative features are more valuable for classification, an easier-to-learn sample should have small class-conditional MD but large class-agnostic MD. The derived RMD is thus an improvement over the class-conditional MD for measuring the sample difficulty, especially when we use pre-trained models that have no direct supervision for downstream classification. We refer to Appendix A for detailed comparisons.

**How well RMD scores the sample difficulty.** Qualitatively, Fig. 1a shows a high-level agreement between human visual perception and RMD-based sample difficulty, and see Appendix C for more visual examples. As shown, hard samples tend to be challenging to classify, as they miss the relevant information or contain ambiguous information. Furthermore, prior works [5, 6] also show that data uncertainty [8] should increase on poor-quality images. We manipulate the sample difficulty by corrupting the input image or alternating the label. Fig. 1b showed that the RMD score increases proportionally with the severity of corruption and label noise, which further shows the effectiveness of RMD in characterizing the hardness and noisiness of samples.

As there is no ground-truth annotation of sample difficulty, we construct the following proxy test for quantitative evaluation. Hard samples are more likely to be misclassified. We therefore use RMD to sort each ImageNet1k validation sample in the descending order of the sample difficulty and group them into subsets of equal size. Fig. 2 shows that an off-the-shelf ImageNet1k classifier (ResNet34 and standard training procedure) performed worst on the hardest data split, and its performance gradually improves as the data split difficulty reduces. This observation indicates an agreement between ResNet34 and RMD regarding what samples are hard and easy. Such an observation also holds for different architectures, see Fig. 7 in Appendix B.

Table 1: Pre-trained Models. We list two variants of CLIP [44] that use ResNet-50 [15] and ViT-Base [10] as the image encoder respectively.

| Model | Pre-trained dataset |
| --- | --- |
| CLIP-ViT-B
CLIP-R50 | 400M image-caption data [44] |
| ViT-B | 14M ImageNet21k (w. labels) [47] |
| MAE-ViT-B | 1.3M ImageNet1k (w./o. labels) [7] |
| DINOv2 | 142M image data [38] |

Fig. 2 also compares four large-scale pre-trained models listed in Table 1 for computing RMD. Interestingly, while MAE-ViT-B [14] is only trained on ImageNet1k (not seeing more data than the downstream classification task), it performs better than ViT-B [10] trained on ImageNet21k.

We hypothesize that self-supervised learning (MAE-ViT-B) plays a positive role compared to supervised learning (ViT-B), as it does not overfit the training classes and instead learn a holistic understanding of the input images beyond low-level image statistics aided by a well-designed loss (e.g., the masking and reconstructing strategy in MAE). [49] also reported that a self-supervised model on ImageNet1k (i.e., SwAV) outperforms supervised models in selecting easy samples to prune for improved training efficiency. Nevertheless, the best-performing model in Fig. 2 is CLIP. The error rate of ResNet34 on the hardest data split rated by CLIP is close to 90%. CLIP learned rich visual concepts from language supervision (cross-modality) and a large amount of data. Thus, it noticeably outperforms the others, which have only seen ImageNet-like data.

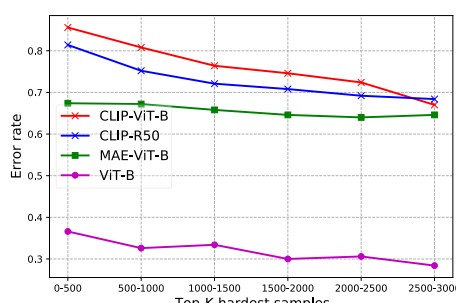

Figure 2: Error rate achieved by ResNet34 (trained on ImageNet1k) on the validation subsets, which respectively contain 500 samples ranked from the $a$th to $b$th hardest. Four different pre-trained models are used for computing RMDs and ranking.

## 4 Difficulty-aware Uncertainty Regularization

While we can clearly observe that a dataset often consists of samples with diverse difficulty levels (e.g., Fig. 1a), the standard training loss formulation is sample difficulty agnostic. In this section, we propose a sample difficulty-aware entropy regularization to improve the training loss, which will be shown to yield more reliable predictions.

Let $f_\theta$ denote the classifier parameterized by a deep neural network, which outputs a conditional probability distribution $p_\theta(y|x)$. Typically, we train $f_\theta$ by minimizing the cross-entropy loss on the training set

$$\ell = E_{(x_i, y_i)} \left\{ -\log \left( f_\theta \left( x_i \right) \left[ y_i \right] \right) \right\}, \tag{8}$$

where $f_\theta(x_i)[y_i]$ refers to the predictive probability of the ground-truth label $y_i$.

It is well known that deep neural networks trained with cross-entropy loss tend to make overconfident predictions. Besides the overfitting reason, it is also because the ground-truth label is typically a one-hot vector which represents the highest confidence regardless of the sample difficulty. Although different regularization-based methods [36, 22, 35] have been proposed to address the issue, they all ignore the difference between easy and hard samples and may assign an inappropriate regularizer for some samples. In view of this, we propose to regularize the cross-entropy loss with a sample difficulty-aware entropic regularizer

$$\ell = E \left\{ -\log \left( f_\theta \left( x_i \right) \left[ y_i \right] \right) - \alpha s(x_i, y_i) \mathcal{H}[f_\theta \left( x_i \right)] \right\}, \tag{9}$$

where $\alpha$ is a hyper-parameter used to control the global strength of entropy regularization and the value range is generally in $(0, 0.5)$. $s(x_i, y_i) \in (0, 1)$ is a normalized sample-specific weighting derived from the RMD-based sample difficulty score (5)

$$s(x_i, y_i) = \frac{\exp \left( \mathcal{RM}(x_i, y_i)/T \right)}{\max_i \{\exp \left( \mathcal{RM}(x_i, y_i)/T \right\} + c}. \tag{10}$$

As the entropic regularizer encourages uncertain predictions, the sample-adaptive weighting $s(x_i, y_i)$, ranging from 0 to 1, is positively proportional to the sample difficulty score $\mathcal{RM}(x_i, y_i)$, where the parameter $c$ equals to a small constant (e.g., 1e-3) ensuring the value range $(0, 1)$. The parameter $T$ is adjustable to control the relative importance among all training data, Fig. 10 in Appendix shows the distribution of $s(\cdot)$ under different $T$.

**Implementation details of sample difficulty score.** Because the difficulty level of a sample may vary depending on how data augmentation is applied, we conduct five random runs to fit Gaussian distributions to eliminate the potential effect. Moreover, we check how the standard augmentation affects the score in Eq. 10, where the numerator can be computed based on each sample with randomly chosen data augmentation, and denominator can be obtained by taking the maximum from the dataset we created to fit the Gaussians in the first step. We do not find data augmentation changes the score too much. Therefore, we can calculate per-sample difficulty score once before training for the sake of saving computing overhead and then incorporate it into the model training using Eq. 9.

# 5 Experiment

We thoroughly compare related uncertainty regularization techniques and sample difficulty measures on various image classification benchmarks. We also verify if our sample difficulty measure conveys important information about the dataset useful to different downstream models. Besides accuracy and uncertainty calibration, the quality of uncertainty estimates is further evaluated on two proxy tasks, i.e., selective classification and out-of-distribution detection.

**Datasets.** CIFAR-10/100 [25], and ImageNet1k [7] are used for multi-class classification training and evaluation. ImageNet-C [17] is used for evaluating calibration under distribution shifts, including blur, noise, and weather-related 16 types of corruptions, at five levels of severity for each type. For OOD detection, in addition to CIFAR-100 vs. CIFAR-10, we use iNaturalist [51] as the near OOD data of ImageNet1k.

**Implementation details.** We use standard data augmentation (i.e., random horizontal flipping and cropping) and SGD with a weight decay of 1e-4 and a momentum of 0.9 for classification training, and report averaged results from five random runs. The default image classifier architecture is ResNet34 [15]. For the baselines, we use the same hyper-parameter setting as recommended in [52]. For the hyper-parameters in our training loss (9), we set $\alpha$ as 0.3 and 0.2 for CIFARs and ImageNet1k, respectively, where $T$ equals 0.7 for all datasets. CLIP-ViT-B [44] is utilized as the default pre-trained model for scoring the sample difficulty.

**Evaluation metrics.** To measure the prediction quality, we report accuracy and expected calibration error (ECE) [37] (equal-mass binning and 15 bin intervals) of the Top-1 prediction. For selective classification and OOD detection, we use three metrics: (1) False positive rate at 95% true positive rate (FPR-95%); (2) Area under the receiver operating characteristic curve (AUROC); (3) Area under the precision-recall curve (AUPR).

## 5.1 Classification accuracy and calibration

Reliable prediction concerns not only the correctness of each prediction but also if the prediction confidence matches the ground-truth accuracy, i.e., calibration of Top-1 prediction.

**I.I.D. setting.** Table 2 reports both accuracy and ECE on the dataset CIFAR-10/100 and ImageNet1k, where the training and evaluation data follow the same distribution. Our method considerably improves both accuracy and ECE over all baselines. It is worth noting that except ER and our method, the others cannot simultaneously improve both ECE and accuracy, trading accuracy for calibration or vice versa. Compared to conventional ER with a constant weighting, our sample difficulty-aware instance-wise weighting leads to much more pronounced gains, e.g., reducing ECE more than $50\%$ on ImageNet1k with an accuracy increase of $+0.43\%$. Furthermore, we also compare with PolyLoss [29], a recent state-of-the-art classification loss. It reformulates the CE loss as a linear combination of polynomial functions and additionally allows flexibly adjusting the polynomial coefficients. In order to improve the accuracy, it has to encourage confident predictions, thus compromising ECE. After introducing the sample difficulty awareness, we avoid facing such a trade-off.

Table 2: The comparison of predictive Top-1 accuracy (%) and ECE (%) on CIFAR-10/-100 and ImageNet1k using ResNet-34. The six baseline methods respectively use the standard cross entropy (CE) loss, CE with label smoothing (LS) [36], focal loss (FL) [35], CE with $L_1$ Norm-based regualarization [22], CE with entropy regularization (ER) [42], and PolyLoss (Poly) [29]. In contrast to our method, all these baseline losses are sample difficulty-agnostic.

|  |  | CE | LS | FL | $L_1$ Norm | ER | Poly | Proposed |
|---|---|---|---|---|---|---|---|---|
| CIFAR-10 | ACC ↑ | 93.79 | 94.47 | 93.82 | 94.51 | 94.34 | 94.66 | **95.67±0.16** |
|  | ECE ↓ | 3.980 | 3.816 | 3.520 | 3.542 | 3.198 | 5.283 | **1.212±0.11** |
| CIFAR-100 | ACC ↑ | 76.79 | 76.48 | 76.74 | 74.32 | 77.32 | 77.39 | **78.58±0.12** |
|  | ECE ↓ | 7.251 | 6.363 | 4.110 | 6.232 | 4.092 | 8.554 | **3.410±0.23** |
| ImageNet1k | ACC ↑ | 73.56 | 73.69 | 72.82 | 73.21 | 73.68 | 74.02 | **74.11±0.10** |
|  | ECE ↓ | 5.301 | 3.994 | 4.901 | 2.625 | 3.720 | 7.882 | **1.602±0.15** |

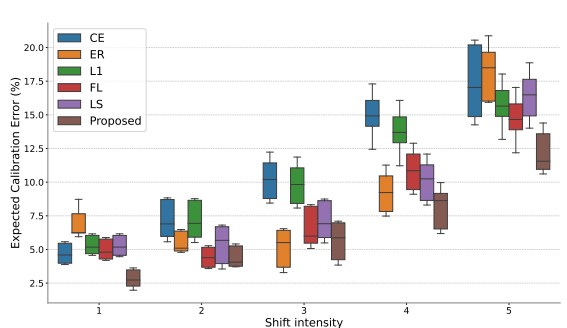

Figure 3: Box plot: ECEs of different methods on ImageNet-C under all types of corruptions with 5 levels of shift intensity. Each box shows a summary of the results of 16 types of shifts.

Figure 4: Accuracy vs. rejection rate, using the predictive entropy as the uncertainty measure to reject predictions on ImageNet1k.

**Improvements across classification models.** If our sample difficulty measure carries useful information about the dataset, it should lead to consistent gains across different classifier architectures (e.g., ResNet, Wide-ResNet and DenseNet). Table 9 in Appendix positively confirms that it benefits the training of small and large models.

**Under distributional shift.** [39] has shown that the quality of predictive uncertainty degrades under distribution shifts, which however are often inevitable in real-world applications. We further evaluate the calibration performance achieved by the ImageNet1k classifier on ImageNet-C [17]. Specifically, Fig. 3 plots the ECE of each method across 16 different types of corruptions at five severity levels, which ranges from the lowest level one to the highest level five. It is prominent that the proposed method surpasses other methods for ECE at various levels of skew. In particular, the gain becomes even larger as the shift intensity increases.

## 5.2 Selective classification

While we do not have the ground-truth predictive uncertainty annotation for each sample, it is natural to expect that a more uncertain prediction should be more likely to be wrong. A critical real-world application of calibrated predictions is to make the model aware of what it does not know. Next, we validate such expectations via selective classification. Namely, the uncertainty measure is used to detect misclassification via thresholding. Rejecting every prediction whose uncertainty measure is above a threshold, we then evaluate the achieved accuracy on the remaining predictions. A high-quality uncertainty measure should improve the accuracy by rejecting misclassifications but also ensure coverage (not trivially rejecting every prediction).

**Misclassification detection.** Table 3 reports the threshold-free metrics, i.e., FPR, AUROC and AUPR, that quantify the success rate of detecting misclassifications based on two uncertainty mea-

sures, i.e., maximum softmax probability (MSP) [18] and entropy. The proposed regularizer is obviously the top-performing one in all metrics on both CIFAR100 and ImageNet1k. The gains are particularly pronounced in FPRs, indicating that the proposed method does not overly trade coverage for accuracy. This is further confirmed in Fig. 4.

Interestingly, our method also reduces the performance gap between the two uncertainty measures, i.e., entropy can deliver similar performance as MSP. Both are scalar measures derived from the same predictive distribution over all classes but carry different statistical properties. While we cannot directly assess the predictive distribution, the fact that different statistics derived from it are all useful is a positive indication.

**Accuracy vs. Coverage.** Fig. 4 plots the accuracy vs. rejection rate on ImageNet1k. Compared to the cross entropy loss (CE) and entropy regularization (ER) with a constant weighting (which is the best baseline in Table 3), our proposal offers much higher selective classification accuracy at the same rejection rate. Moreover, the gain (orange area) increases along with the rejection rate, indicating the effectiveness of our method. Without sample difficulty-aware weighting, the accuracy gain of ER over CE vanishes on ImageNet1k.

In Appendix D, we also compare the predictive entropy and confidence of different methods for wrong predictions, see Fig 11. The proposed method achieves the lowest predictive confidence and highest entropy among all methods for misclassified samples, which further verifies the reliable uncertainty estimation of the proposed regularizer.

Table 3: The comparison of misclassification detection performance (%).

| Dataset | Method | FPR-95%↓ | AUROC ↑ | AUPR ↑ |
|---|---|---|---|---|
| | | MSP / Entropy | | |
| | CE | 45.21/47.44 | 86.29/86.58 | 94.70/95.47 |
| | LS | 49.19/49.71 | 86.00/85.82 | 94.31/94.19 |
| C100 | FL | 47.30/48.84 | 85.90/84.72 | 94.90/94.50 |
| | $L_1$ | 48.11/48.40 | 85.52/85.31 | 94.05/93.90 |
| | ER | 45.40/46.88 | 86.32/85.69 | 95.26/95.20 |
| | CRL | 44.80/46.08 | 86.67/85.98 | 95.55/95.49 |
| | Ours | **42.71/43.22** | **87.50/87.03** | **96.10/96.02** |
| | CE | 46.93/48.94 | 86.03/85.02 | 94.22/93.75 |
| | LS | 52.12/61.52 | 85.28/81.03 | 93.86/91.98 |
| ImageNet1k | FL | 48.54/51.10 | 85.67/83.05 | 94.08/93.03 |
| | $L_1$ | 47.19/48.58 | 86.19/84.58 | 94.39/93.82 |
| | ER | 46.98/48.85 | 86.15/84.19 | 94.38/93.82 |
| | CRL | 46.03/48.01 | 86.11/84.33 | 94.41/93.89 |
| | Ours | **45.69/46.72** | **86.53/85.23** | **94.76/94.31** |

Table 4: The comparison of near OOD detection performance (%).

| $\mathcal{D}_{in}/\mathcal{D}_{out}$ | Method | FPR-95%↓ | AUROC ↑ | AUPR ↑ |
|---|---|---|---|---|
| | | MaxLogit / Entropy | | |
| | CE | 60.19/60.10 | 78.42/78.91 | 79.02/80.19 |
| | LS | 69.51/70.08 | 77.14/77.31 | 75.92/75.89 |
| C100/C10 | FL | 61.02/61.33 | 78.79/79.36 | 80.22/80.22 |
| | $L_1$ | 61.58/61.60 | 76.52/76.58 | 79.19/79.21 |
| | ER | 59.52/59.92 | 79.22/79.46 | 81.01/81.22 |
| | CRL | 58.13/58.54 | 79.91/80.13 | 81.75/81.89 |
| | Ours | **55.48/55.60** | **80.20/80.72** | **82.51/82.84** |
| | CE | 36.06/40.33 | 89.82/89.02 | 97.61/97.40 |
| | LS | 37.56/41.22 | 89.10/89.35 | 97.12/97.30 |
| ImageNet1k / | FL | 36.50/36.32 | 90.03/90.25 | 97.76/97.80 |
| iNaturalist | $L_1$ | 38.32/46.03 | 88.90/88.29 | 97.53/97.27 |
| | ER | 36.00/35.11 | 89.86/90.04 | 97.75/97.43 |
| | CRL | 35.07/34.65 | 90.11/90.32 | 97.96/97.81 |
| | Ours | **32.17/34.19** | **91.03/90.65** | **98.03/97.99** |

## 5.3 Out-of-distribution detection (OOD)

In an open-world context, test samples can be drawn from any distribution. A reliable uncertainty estimate should be high on test samples when they deviate from the training distribution. In light of this, we verify the performance of the proposed method on near OOD detection (i.e., CIFAR-100 vs CIFAR-10 and ImageNet1k vs iNaturalist), which is a more difficult setting than far OOD tasks (e.g., CIFAR100 vs SVHN and CIFAR10 vs MNIST) [12].

As our goal is to evaluate the quality of uncertainty rather than specifically solving the OOD detection task, we leverage MaxLogit and predictive entropy as the OOD score. [16] showed MaxLogit to be more effective than MSP for OOD detection when dealing with a large number of training classes, e.g., ImageNet1k. Table 4 shows that the uncertainty measures improved by our method lead to improvements in OOD detection, indicating its effectiveness for reliable predictions.

## 5.4 The robustness of hyperparameter $\alpha$

We analyze the effects of hyper-parameters: $\alpha$ in ER and the proposed method on the predictive performance, and Table 5 reports the comparison results of different $\alpha$ on ImageNet1k. Compared to ER, the proposed method is less sensitive to the choice of owing to sample adaptive weighting, whereas ER penalizes confident predictions regardless of easy or hard samples. Oftentimes, a large

Table 5: The comparison for ACC and ECE under different $\alpha$ on ImageNet1k. **Bold** indicates the results from the chosen hyperparameter.

| $\alpha$ | | 0 | 0.05 | 0.10 | 0.15 | 0.20 | 0.25 | 0.30 |
|---|---|---|---|---|---|---|---|---|
| ER | ACC | 73.56 | 73.59 | **73.68** | 73.76 | 73.78 | 73.81 | 73.88 |
| | ECE | 5.301 | 3.308 | **3.720** | 10.19 | 15.24 | 25.22 | 33.26 |
| Proposed | ACC | 73.56 | 73.89 | 73.91 | 74.08 | **74.11** | 74.13 | 73.98 |
| | ECE | 5.301 | 1.855 | 1.851 | 1.596 | **1.602** | 1.612 | 1.667 |

portion of data is not difficult to classify, but the equal penalty on every sample results in the model becoming underconfident, resulting to poor ECEs.

### 5.5 Comparison of different sample difficulty measures

Finally, we compare different sample difficulty measures for reliable prediction, showing the superiority of using large-scale pre-trained models and the derived RMD score.

**CLIP is the top-performing model for sample difficulty quantification.** In addition to Fig. 2, Table 6 compares the pre-trained models listed in Table 1 in terms of accuracy and ECE. All pre-trained models improve the baseline (using the standard CE loss), especially in ECE. CLIP outperforms ViT-B/MAE-ViT-B, thanks to a large amount of diverse multi-modal pre-trained data (i.e., learning the visual concept from language supervision). Additionally, the performance of using recent DINOv2 [38] as a pre-trained model to quantify sample difficulty also surpasses ViT-B/MAE-ViT-B. While ImageNet21k is larger than ImageNet1k, self-supervised learning adopted by MAE-ViT-B reduces overfitting the training classes, thus resulting in better performance than ViT-B (which is trained supervisedly).

Table 6: The comparison of different pre-trained models for accuracy and ECE on ImageNet1k.

| | ViT-B | MAE-ViT-B | DINOv2 | CLIP-R50 | CLIP-ViT-B |
|---|---|---|---|---|---|
| ACC $\uparrow$ | 73.59 | 73.81 | 74.01 | 74.05 | **74.11** |
| ECE $\downarrow$ | 2.701 | 1.925 | 1.773 | 1.882 | **1.602** |

**CLIP outperforms supervised models on the task dataset.** To further confirm the benefits from large-scale pre-training models, we compare them with supervised models on the task dataset (i.e., ImageNet1k). We use three models (i.e., ResNet34/50/101) trained with 90 epochs on ImageNet1k to measure the sample difficulty. We did not find it beneficial to use earlier-terminated models. The classifier model is always ResNet34-based. Besides RMD, we also exploit the training loss of each sample for sample difficulty quantification. Table 10 in Appendix shows that they all lead to better ECEs than ER with constant weighting (see Table 2). Compared to the pre-trained models in Table 6, they are slightly worse than ViT-B in ECE. Without exposure to the 1k classes, MAE-ViT-B achieves a noticeably better ECE. The pronounced improvements in both accuracy and ECE shown in the last column of Table 10 are enabled by CLIP.

**Effectiveness of RMD.** We resorted to density estimation for quantifying sample difficulty and derived the RMD score. Alternatively, [50] proposed to use the frequency of correct predictions during training epochs. Based on that, [34] proposed a new loss termed correctness ranking loss (CRL). It aims to ensure that the confidence-based ranking between the two samples should be consistent with their frequency-based ranking. Table 7 shows our method scales better with larger datasets. Our RMD score is derived from the whole data distribution, thus considering all samples at once rather than a random pair at each iteration. The ranking between the two samples also changes over training, while RMD is constant, thus ensuring a stable training objective.

Next, we compare different distance measures in the feature space of CLIP-ViT-B, i.e., K-means, MD and RMD. Table 8 confirms the superiority of RMD in measuring the sample difficulty and ranking the training samples.

Table 7: The comparison of RMD and correctness ranking loss (CRL) [34] in regularizing the prediction confidence.

| Method | | C10 | C100 | ImageNet1k |
|---|---|---|---|---|
| CRL | ACC ↑ | 94.06 | 76.71 | 73.60 |
| | ECE ↓ | **0.957** | 3.877 | 2.272 |
| Proposed | ACC ↑ | **95.67** | **78.58** | **74.11** |
| | ECE ↓ | 1.212 | **3.410** | **1.602** |

Table 8: The comparison of different sample difficulty measures for predictive Top-1 accuracy (%) and ECE (%) on ImageNet1k.

| | K-means | MD | RMD |
|---|---|---|---|
| ACC ↑ | 73.78 | 73.71 | **74.11** |
| ECE ↓ | 2.241 | 2.125 | **1.613** |

## 6 Conclusions

This work introduced a new application of pre-trained models, exploiting them to improve the calibration and quality of uncertainty quantification of the downstream model. Profiting from rich large-scale (or even multi-modal) datasets and self-supervised learning for pre-training, pre-trained models do not overfit the downstream training data. Hence, we derive a relative Mahalanobis distance (RMD) via the Gaussian modeling in the feature space of pre-trained models to measure the sample difficulty and leverage such information to penalize overconfident predictions adaptively. We perform extensive experiments to verify our method's effectiveness, showing that the proposed method can improve prediction performance and uncertainty quantification simultaneously.

In the future, the proposed method may promote more approaches to explore the potential of large-scale pre-trained models and exploit them to enhance the reliability and robustness of the downstream model. For example, for the medical domain, MedCLIP [53] can be an interesting alternative to CLIP/DINOv2 for practicing our method. Besides, conjoining accuracy and calibration is vital for the practical deployment of the model, so we hope our method could bridge this gap. Furthermore, many new large-scale models (including CLIP already) have text-language alignment. In our method, we do not explore the language part yet, however, it would be interesting to use the text encoder to "explain" the hard/easy samples in human language. One step further, if the pre-trained models can also synthesize (here CLIPs cannot), we can augment the training with additional easy or hard samples to further boost the performance.

**Potential Limitations.** Pre-trained models may have their own limitations. For example, CLIP is not straightforwardly suitable for medical domain data, as medical images can be OOD (out-of-distribution) to CLIP. If targeting chest x-rays, we need to switch from CLIP to MedCLIP [53] as a pre-trained model to compute the sample difficulty. In this work, CLIP is the chosen pre-trained model, as it is well-suited for handling natural images, which are the basis of the targeted benchmarks. Besides, the calculation of sample difficulty introduces an additional computational overhead although we find it very affordable.

## Acknowledgments

This work was supported by NSFC Projects (Nos. 62061136001, 92248303, 62106123, 61972224), Natural Science Foundation of Shanghai (No. 23ZR1428700) and the Key Research and Development Program of Shandong Province, China (No. 2023CXGC010112), Tsinghua Institute for Guo Qiang, and the High Performance Computing Center, Tsinghua University. J.Z is also supported by the XPlorer Prize.

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

# A The comparison of MD and RMD for measuring the sample difficulty

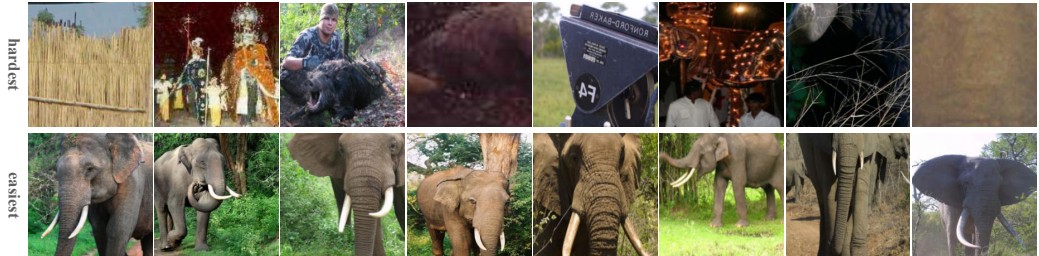

(a) Visualization of the Top-8 hardest samples (top row) and Top-8 easiest samples (bottom row) in ImageNet (class Tusker) which are ranked by means of the CLIP-VIT-B-based RMD score.

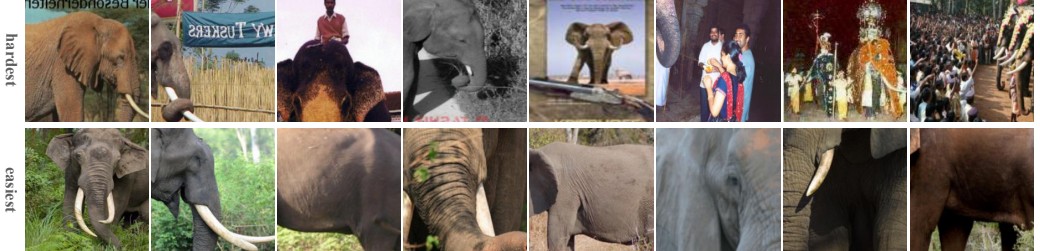

(b) Visualization of the Top-8 hardest samples (top row) and Top-8 easiest samples (bottom row) in ImageNet (class Tusker) which are ranked by means of the CLIP-VIT-B-based MD score.

Figure 5: Visualization of the Top-k hardest and easiest samples in ImageNet (class Tusker) which are ranked by RMD and MD scores. In contrast to the MD score, the easy and hard samples measured by RMD are more accurate than those by MD.

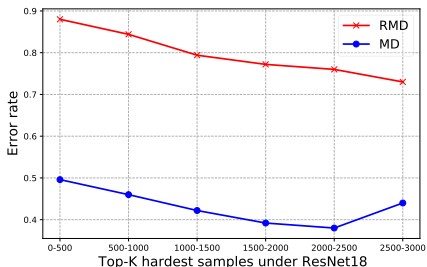

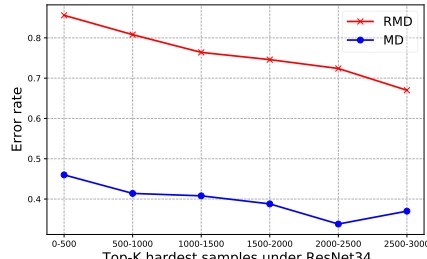

(a) Error rate achieved by ResNet18 (trained on ImageNet) on the validation subsets, which respectively contain 500 samples ranked from the $a$th to $b$th hardest.

(b) Error rate achieved by ResNet34 (trained on ImageNet) on the validation subsets, which respectively contain 500 samples ranked from the $a$th to $b$th hardest.

Figure 6: The performance comparison of RMD and MD for characterizing Top-K hardest samples.

In Fig. 5, we further compare Top-k hardest and easiest samples that are ranked by RMD (Fig. 5a) and MD (Fig. 5b) scores respectively. We can see that hard and easy samples characterized by RMD are more accurate than those characterized by MD, and there is a high-level agreement between human visual perception and RMD-based sample difficulty. Moreover, we quantitatively compare the performance of RMD and MD for characterizing Top-K hardest samples in Fig 6. We can observe that the error rate of ResNet18 and ResNet34 on the hardest data split rated by RMD is close to 90%, which significantly suppresses the performance of MD. Therefore, the derived RMD in this paper is an improvement over the class-conditional MD for measuring the sample difficulty.

# B  Additional comparisons for different pre-trained models

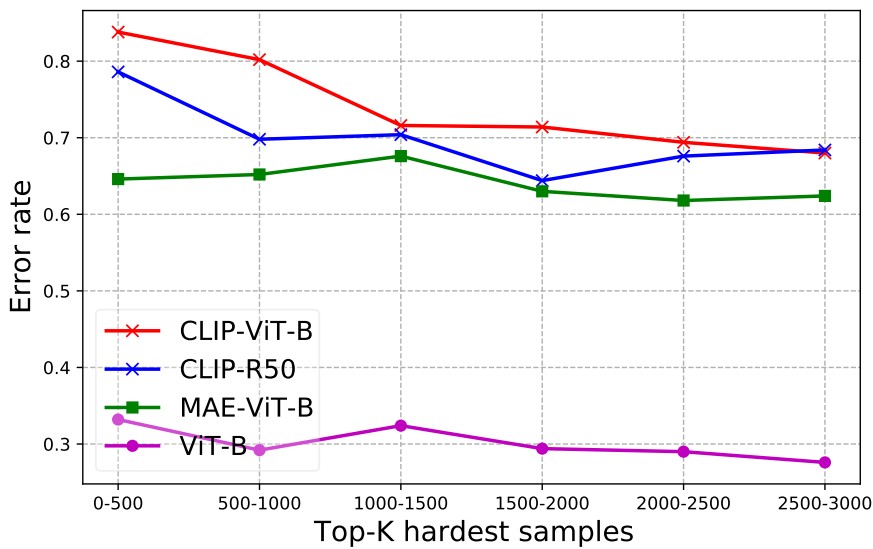

Figure 7: Error rate achieved by DenseNet121 (trained on ImageNet) on the validation subsets, which respectively contain $500$ samples ranked from the $a$th to $b$th hardest. Four different pre-trained models are used for computing RMDs and ranking. They all show the same trend, i.e., the error rate reduces along with the sample difficulty. However, ViT-B supervisedly trained on ImageNet21k performed much worse than the others.

# C  More hard and easy samples ranked by RMD

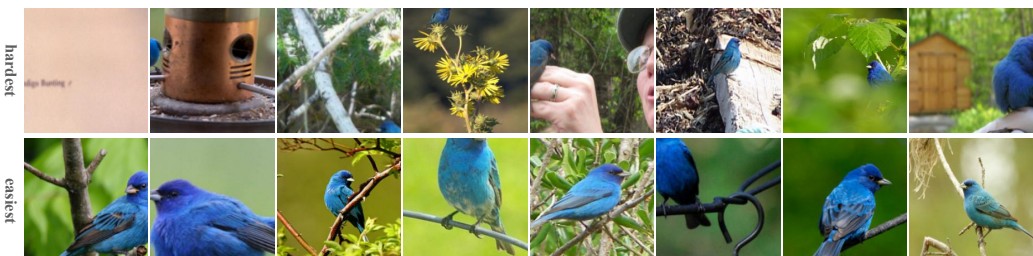

Figure 8: Visualization of the Top-8 hardest samples (top row) and Top-8 easiest samples (bottom row) in ImageNet (class indigo bird) which are ranked by means of the CLIP-VIT-B-based RMD score.

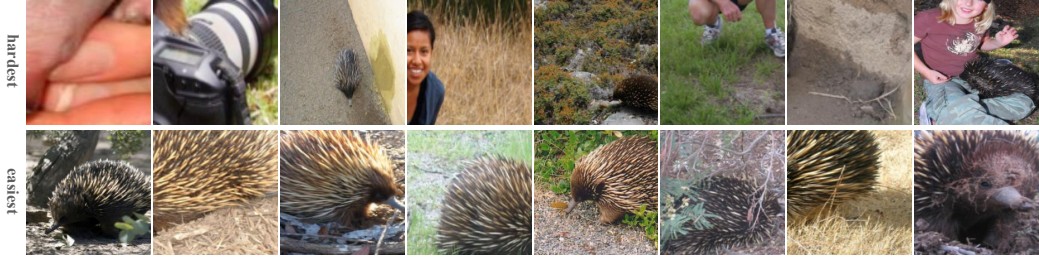

Figure 9: Visualization of the Top-8 hardest samples (top row) and Top-8 easiest samples (bottom row) in ImageNet (class echidna) which are ranked by means of the CLIP-VIT-B-based RMD score.

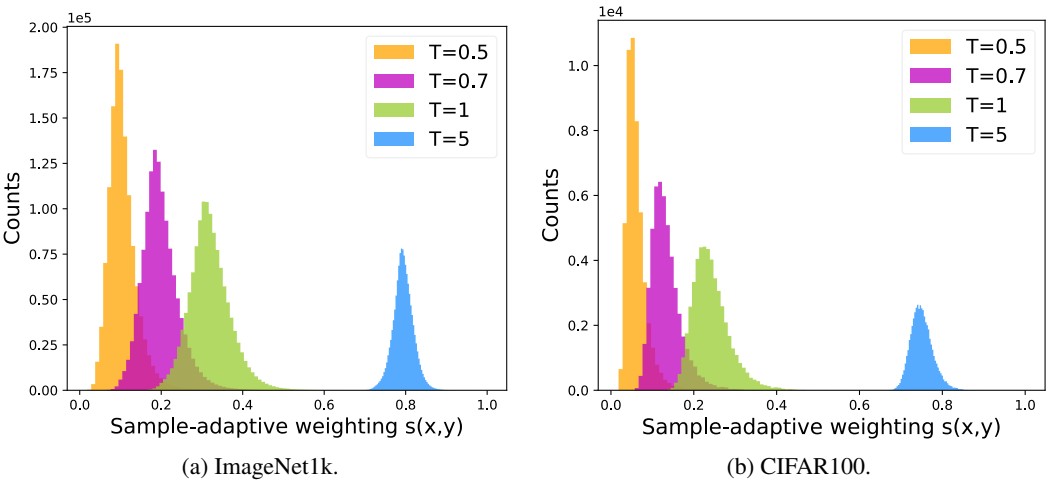

(a) ImageNet1k.

(b) CIFAR100.

Figure 10: Histograms of $s(x_i, y_i)$ at different $T$.

# D More experimental results

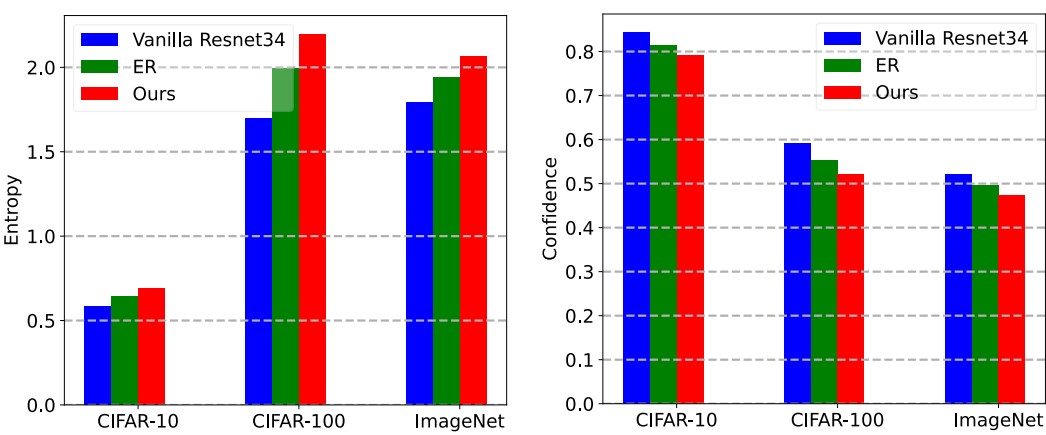

(a) Predictive entropy of misclassified samples on CI-FAR and ImageNet datasets.

(b) Predictive confidence of misclassified samples on CIFAR and ImageNet datasets.

Figure 11: Predictive entropy and confidence of misclassified samples for different methods on CIFAR and ImageNet datasets.

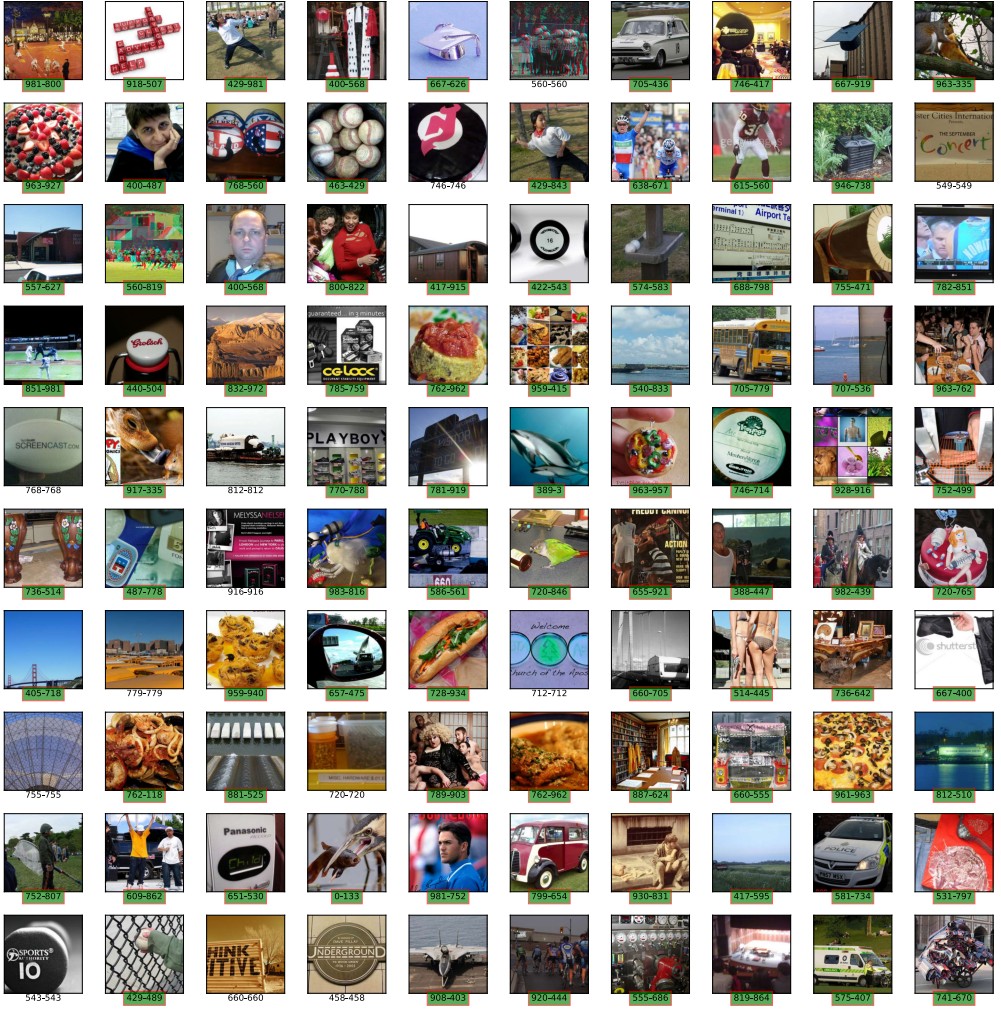

Figure 12: Test samples with predictions corresponding to the Top-100 relative Mahalanobis distance. The text box ("x-x") with green shade represents a misclassification. The first number indicates the label class, and the second number indicates the predictive class.

Table 9: The comparison of various architectures for predictive Top-1 accuracy (%) and ECE (%) on ImageNet1k.

| Arch. | | CE | ER | Proposed |
|---|---|---|---|---|
| ResNet18 | ACC ↑ | 70.46 | 70.59 | **70.82** |
| | ECE ↓ | 4.354 | 2.773 | **1.554** |
| ResNet34 | ACC ↑ | 73.56 | 73.68 | **74.11** |
| | ECE ↓ | 5.301 | 3.720 | **1.602** |
| ResNet50 | ACC ↑ | 76.08 | 76.11 | **76.59** |
| | ECE ↓ | 3.661 | 3.212 | **1.671** |
| DenseNet121 | ACC ↑ | 75.60 | 75.73 | **75.99** |
| | ECE ↓ | 3.963 | 3.010 | **1.613** |
| WRN50x2 | ACC ↑ | 76.79 | 76.81 | **77.23** |
| | ECE ↓ | 4.754 | 2.957 | **1.855** |

Table 10: The comparison of different model-based measures for predictive Top-1 accuracy (%) and ECE (%) on ImageNet1k. Compared to the three ResNets, "$\Delta$" denotes the averaged gain achieved by CLIP-ViT-B in Table 6.

| Measures | | ResNet34 | ResNet50 | ResNet101 | $\Delta$ |
|----------|-----------------|----------|----------|-----------|----------|
| RMD | ACC $\uparrow$ | 73.73 | 73.78 | 73.88 | +0.31 |
| | ECE $\downarrow$ | 3.298 | 2.996 | 2.882 | −1.44 |
| Loss | ACC $\uparrow$ | 73.58 | 73.61 | 73.75 | +0.46 |
| | ECE $\downarrow$ | 3.624 | 2.997 | 2.783 | −1.52 |

