# OpenReview forum: "Learning Sample Difficulty from Pre-trained Models for Reliable Prediction"
_NeurIPS.cc/2023/Conference — NeurIPS 2023 poster_

### Official Review · Reviewer_93FC · 2023-07-05

**Soundness:** 3 good
**Presentation:** 3 good
**Contribution:** 3 good
**Rating:** 7
**Confidence:** 4

**Summary:**

It is known that state-of-art deep-network based machine learning models are poorly calibrated. It is also somewhat widely known that the poor calibration is because we do not model data uncertainty while training.
This work proposes to use CLIP pretrained model for estimating the sample difficulty, which when further used for modeling data uncertainty is shown to yield better calibrated models.

Relative Maholonibis distance and leveraging a multimodal pretrained model such as CLIP for estimating the sample difficulty is the core contribution of this work.

**Strengths:**

- Convincing experimental study. All the components of their proposed method: CLIP, RMD, alpha (hyperparameter) are well argued.
- Strong evaluation. The proposed method is evaluated with positive results on various downstream applications: outlier detection, selective classification, ECE on OOD.
- Writing is easy to follow.

**Weaknesses:**

*Limited originality*: I enjoyed reading the paper but it cuts very close to the existing work.
As mentioned in the paper, several previous methods attempted example weighting during training [39] based on their difficulty, and RMD is proposed and used earlier for outlier detection [45] (citation numbers from their paper).
As I see it, the originality of the paper lies in using RMD and pretrained models for sample weighting. Although the paper is technically strong, I am somewhat unimpressed by its originality.

*Universality of CLIP*. The proposed measure of sample difficulty is in essence how well CLIP can identify different classes.
Although CLIP may have been trained on large and diverse data, it surely has its limits.
We may then also need an additional measure that informs how difficult it is for the CLIP to provide a sample diffcilty for an example.
For instance, the proposed measure most likely cannot be useful on a chest xray dataset for a downstream task of classifying pneumonia or normal.
Comments on how such scenarios can be addressed are missing from the paper.

**Questions:**

- Is the base model used for training on downstream models (which is mentioned as ResNet34) is pretrained?
- What happens when CLIP itself is used as the base model?
- Can some justification be provided on why we see better corruption robustness in ECE in Fig. 3?

I would also appreciate if the authors can respond, if they can, to the weaknesses.

*Minor comments*.
- L171-182. I found the writing somewhat superfluous since we only care to see that the error rate dips as we move towards right in Fig 2. Perhaps there is no need to comment about why ViT is the best etc. Also, there seems to be some confusion between the text and Fig.2. Text interprets the y axis as accuracy while it is error rate in the figure.
- L310: last sentence seems incomplete.


**Limitations:**

Yes.

Edit after rebuttal: Please see my comment in the thread below.

---

> ### Author Rebuttal · Authors · 2023-08-09
>
> Thank you for appreciating our convincing experimental study and strong evaluation. We address the detailed concerns below. We hope that you may find our response satisfactory and raise your score accordingly.
>
> **Q1: Limited originality, such as example weighting and RMD:** We understand your viewpoint. Although our method builds upon the established concepts of ER and RMD, it makes a simple modification that greatly improves upon classical regularization techniques. Furthermore, our work is the first to leverage pre-trained models for sample difficulty annotation and demonstrate an effective way to use sample difficulty for reliable prediction (i.e., simultaneously improving accuracy and uncertainty calibration). We believe the simplicity of the proposed approach will make it more likely that this work is adopted or studied in the community.
>
> **Q2: Limits of CLIP itself, e.g., cannot be useful on a chest xray dataset, and the need of a usability indicator:** Yes, CLIP has its own limits. It is likely that CLIP is not straightforwardly suitable for medical domain data. Nevertheless, our method itself can work with different pre-trained models. For instance, if targeting chest xray, our method can be practiced as computing RMD in the feature space of MedCLIP [*1] for measuring the sample difficulty. In this work, CLIP is the chosen pre-trained model, as it is well-suited for handling natural images, which are the basis of the targeted benchmarks. We perform a new experiment during the rebuttal, using recent DINOv2 [*2] as a pre-trained model to quantify sample difficulty, and report ACC and ECE on CIFAR-10/100 in Table Ⅱ of the attached rebuttal PDF in our global response. As shown, our method based on DINOv2 also shows superior performance similar to CLIP and significantly outperforms the baselines, confirming the compatibility of our method with different pre-trained models.
>
> Moreover, a simple way to assess the effectiveness of a pre-trained model within a specific domain can be achieved by examining the zero-shot classification performance. For instance, if CLIP cannot deliver strong zero-shot generalization on a chest xray classification, then we should choose a different pre-trained model or adapt CLIP for that domain first.
>
> Thank you for your suggestion. We will include the above discussion upon revision.
>
> Reference:
>
> [*1] MedCLIP: Contrastive Learning from Unpaired Medical Images and Text, Wang et al., EMNLP 2022
>
> [*2] DINOv2: Learning Robust Visual Features without Supervision, Oquab et al., arXiv:2304.07193
>
> **Q3: Is the base model used for training on downstream models (which is mentioned as ResNet34) is pre-trained?** We understand here your 'base model' refers to the model that is used for training the classifier, e.g., ResNet34. If this understanding is correct, then the base model is randomly initialized and trained from scratch on downstream tasks. For the models that are used for sample difficulty measurement, they are pre-trained with the datasets documented in Table 8 in Appendix.
>
> **Q4: What happens when CLIP itself is used as the base model?** Using CLIP as the base model, fine-tuning it on the downstream dataset is necessary to boost the accuracy. However, fine-tuning has the forgetting issue, as the downstream dataset is usually smaller than the pre-training dataset. Furthermore, relying on the single ground-truth label, fine-tuning would still lead to over-confident predictions. Therefore, there is no obvious evidence that it has the potential to outperform our method. Plus, our method is applicable to any base model architecture, whereas fine-tuning CLIP is limited to a small set of architectures.
>
> Besides, we also report ACC and ECE of CLIP-RN50 fine-tuned with a linear classifier (i.e., only fine-tuning the last linear layer to alleviate the forgetting/overfitting issue and save computational cost) on CIFAR10/100 in Table A. It is prominent that fine-tuned CLIP achieves comparable accuracy but exhibits poor calibration performance (i.e., higher ECE) .
>
> Tab.A: The comparison of fine-tuned CLIP and our method on CIFAR-10/100.
> | Method |     | CIFAR-10 | CIFAR-100 |
> |-------------------|-----|----------|-----------|
> | Fine-tuned CLIP   | ACC | 94.32    | 77.16     |
> |                   | ECE | 3.812    | 6.358     |
> | Ours              | ACC | **95.67**    | **78.58**     |
> |                   | ECE | **1.212**    | **3.410**     |
>
> **Q5: Can some justification be provided on why we see better corruption robustness in ECE in Fig. 3?** Our method can alleviate overconfidence and improve uncertainty calibration (lower ECE) owing to the proposed sample adaptive weighting. Therefore, the proposed method has the ability to make conservative predictions (low ECE) for OOD inputs (i.e., images under different levels of corruption). It has also been observed in the literature, e.g., [*2], that models with good uncertainty estimation quality are also more robust under data shifts, e.g., image corruptions.
>
> Reference:
>
> [*3] Can You Trust Your Model's Uncertainty? Evaluating Predictive Uncertainty Under Dataset Shift, Yaniv et al., NeurIPS 2019
>
> **Q6: Editing or writing errors:** Thank you for pointing out the editing error, and we will correct it upon revision.

---

> > ### Comment · Reviewer_93FC · 2023-08-12
> >
> > Thank you very much fo detailed response and additional experiments.
> >
> > All my concerns are well addressed. Please make sure to include eveluation proposals to test suitability of a pretrained model to a domain. Although the concern on originality still remains, this work is a thorough study that I would like to see presented at the conference. I am increasing my score accordingly.

---

> > > ### Author Response · Authors · 2023-08-13
> > > **Thank you very much for increasing the score!**
> > >
> > > We are pleased to know that Reviewer 93FC founds our rebuttal satisfactory and increases the score accordingly. We will include eveluation proposals to test suitability of a pretrained model to a domain in the final version.

---

### Official Review · Reviewer_Lxfs · 2023-07-06

**Soundness:** 3 good
**Presentation:** 4 excellent
**Contribution:** 3 good
**Rating:** 6
**Confidence:** 4

**Summary:**

To address the over-confidence problem of the uncertainty estimation in deep learning, this paper for the first time proposes to use the pre-trained large models to estimate the learning difficulty of each sample. Then, the estimated sample difficulty information is embedded in the final loss for training deep models. The authors evaluate the proposed framework on different datasets and with different model architectures, showing that it improves uncertainty calibration on both ID and OOD settings.

**Strengths:**

1. The studied problem namely uncertainty calibration is important in modern deep learning. The presentation of this paper is good and the paper is easy to follow.

2. Different from some previous works that use pre-trained large models as the base models to fine-tune, this paper proposes to use large models as the tools to estimate sample difficulty, which makes the training process flexible since any model architecture can be used in the training.

3. I like the idea of using sample difficulty as a metric to guide the modeling of uncertainty. And based on the authors' observation, large models (especially the models learned with cross-modality) seems give more accurate estimation of sample difficulty.

**Weaknesses:**

1. The proposed distance metric RMD in Eq. (5) is shown useful than MD in Fig. 6, but backs of the theoretic supports.

2. Two new hyperparameters ($\alpha$ and T) are induced in the final loss function (9), used as the weighting factors of the entropy penalty term, which act like the factor used in label smoothing and may reduce the robustness of the proposed method.

3. I wonder if the large model necessary for the estimation of sample difficulty? There must be some other metrics for sample difficulty, so if other metrics also yield good performance with the regularization-based loss function (9). Maybe some ablation studies can be conducted for this point.


**Questions:**

1. If no theoretic support could be added, could you explain more about why RMD is better than MD.

2. Maybe experiments of hyperparameter study can be added for the second weakness.


**Limitations:**

N/A.

---

> ### Author Rebuttal · Authors · 2023-08-09
>
> Thank you for appreciating our essential problems and new contributions and providing valuable comments. We address the detailed concerns below. We hope that you may find our response satisfactory and raise your score accordingly.
>
> **Q1: why RMD is better than MD?** The key difference between RMD and MD lies in the relative distance calculation. MD scores the sample difficulty based on the distance towards the class-specific mean mode, whereas RMD examines the relative distances towards two properly chosen mean modes, i.e., class-agnostic and class-specific mean. The relative distance reflects not only if the feature is typical (close to the class-specific mean), but also if it is discriminative (far from the class-agnostic mean). MD mostly concerns the first aspect, thus performs worse than RMD, esp. when taking care of fine-grained classification benchmarks such as ImageNet1k. In ImageNet1k, many classes could share some common features, e.g., there are many sub-classes of elephants in ImageNet1k and they all look alike in some aspects. To confidently classify one elephant as Tusker, it must have typical features that are also discriminative enough from the rest of the classes. The metrical difference using RMD can filter out the influence from shared typical features, i.e., features close to the class-agnostic mean mode. We will make it clearer in the next version. We also add an illustrative example in Figure I of the attached rebuttal PDF in our global response, which hopefully visualizes well the above description.
>
> **Q2: Robustness of hyperparameter $\alpha$ and $T$, ablation results:** In our experiments, we did not experience hard tuning scenarios. Being part of our ablation study on ImageNet1k classification, we selected the weighting coefficient $\alpha$ from { 0.05, 0.10, 0.15, 0.20, 0.25, 0.30 }. As shown in Table 4, the proposed method is less sensitive to the choice of $\alpha$ than the baseline ER, owing to our sample adaptive weighting. As for the temperature $T$, it is set to 0.7 for all datasets. Hence, the proposed method does not rely on time-consuming hyper-parameters tuning.
>
> **Q3: Is the large model necessary to estimate sample difficulty? Ablation studies about different metrics for sample difficulty:** To verify the benefit of using large models, we compared them with smaller ResNet34/50/101 models that were trained on the task dataset in Table 10 in Appendix. We find large models beneficial to estimate the sample difficulty. As they were trained on large-scale datasets with high sample diversities in many dimensions, they can learn to preserve and structure richer semantic features of the training samples than models only exposed to the training set that is commonly used at a smaller scale. Hence, large pre-trained models have a representative feature space, which can be used to quantify more accurate sample difficulty scores. Besides CLIP, we additionally use DINOv2 [*1] during the rebuttal, which is also a large-scale pre-trained model. The ACC and ECE gains are comparable with CLIP, significantly outperforming the baselines.
>
> Moreover, we have compared different metrics (K-means and MD) for sample difficulty in Table 7, which confirms the superiority of RMD in measuring the sample difficulty. We also compared our method and CRL (a baseline that uses the frequency of correct predictions during training epochs as a sample difficulty metric) in Table 6, and we can see that large pre-trained models still perform better.
>
> Reference:
>
> [*1] DINOv2: Learning Robust Visual Features without Supervision, Oquab et al., arXiv:2304.07193

---

> > ### Comment · Reviewer_Lxfs · 2023-08-22
> > **Thanks for the responses.**
> >
> > Thank you for your responses. I have thoroughly reviewed the feedback, including the additional experimental results and the illustrative figure provided in the attached rebuttal PDF.
> >
> > The results from the ablation studies align with the stated claims. However, I still maintain concerns regarding the issue of hyperparameters. Given the various choices available for the combination of $\alpha$ and $T$, and the uncertainty regarding the method's sensitivity to $T$, it remains challenging to definitively assert that larger models consistently outperform others. As the utilization of larger models stands as a pivotal contribution within this paper, I am maintaining my score.

---

> > > ### Author Response · Authors · 2023-08-22
> > > **Thank you for your feedback**
> > >
> > > Thanks for reviewing our rebuttal and providing feedback. We understand your viewpoint, but have a different perspective on "the issue of hyperparameters”. The utilization of $\alpha$ is common in various regularization-based methods, and experimental results in Table 4 have verified that our method does not rely on time-consuming hyperparameters tuning. Besides, "temperature" is a widely-used hyperparameter in many loss functions, such as uncertainty calibration, knowledge distillation, and contrastive loss. We can flexibly control the relative importance among all training data by tuning parameter “T”, and a fixed “T” for all datasets can also work well in our method. Hence, we believe that both hyperparameters are not limiting factors of our method.

---

### Official Review · Reviewer_YeMZ · 2023-07-06

**Soundness:** 3 good
**Presentation:** 3 good
**Contribution:** 3 good
**Rating:** 6
**Confidence:** 4

**Summary:**

The authors propose to improve model calibration by leveraging information about a sample's difficulty. To do this, they cluster samples using embeddings obtained from large, pre-trained models, and then use a sample's distance to samples from the same class as proxy for difficulty. They show improvements on imagenet in calibration and slight improvements on accuracy.

**Strengths:**

Originality: The approach is effectively a form of adaptive entropy regularization. The novelty is that the strength of the regularization depends on the "difficulty" of the samples. The approach makes sense and is (to the best of my knowlege) novel.

Quality: Theoretical analysis is a bit lacking. Empirical results make sense and show a decent improvement in calibration scores. As such, the method does make sense.

Clarity: Some choices the paper makes are unclear to me, both in terms of overall method design (why focus so much on RMD?) and in details (e.g. formulation of Eq. 10). See "details" for more. As far as language use goes, the paper is easily understandable.

Significance: Results show that the idea works. ECE decreases, and the qualitative examples shown in the figures/supplementary make sense to me.

**Weaknesses:**

* The main idea of the paper (estimate difficulty of samples on large pre-trained model, weight regularization with that) is a bit obscured by the fact that the authors make a big fuzz about using the "Relative Mahaloni Distance" (RMD), even though Table 7 shows that even doing simple K-Means clustering works as well (even better than normal Entropy Regularization, if one compares with Table 4). The paper's main idea would be much more obvious if the paper first established that the idea works even using a simple clustering like K-Means, and then shows that RMD is a sensible improvement upon this.  As such, I don't really understand why RMD was chosen: The paper also never goes into explaining WHY RMD beats K-Means, or why doing Mahaloni Distance (MD) does not work as well. The authors merely say that RMD worked well for near-OOD detection.


**Questions:**

* Why take the maximum over i in Eq. 10?

* Table4: misses a description of what the columns represent. The text says that they are different regularization strenghts, but this should also be made explicit in the table itself.

* Authors state that all results are averages, but never talk about the variances between runs. It would be nice to know if error bars overlap.

* What is the runtime of calculating RMD? It would be nice to know if this method is applicable in practice or is prohibitively expensive.

* How well does the Gaussian assumption fit? An exploration of how well the RMD describes the actual representation space would be interesting.

**Limitations:**

Authors do not adress limitations of their approach, and I'd encourage them to add a discussion of those!

---

> ### Author Rebuttal · Authors · 2023-08-09
>
> We thank you for finding our work novel and clarifying empirical results. We address the detailed concerns below. We hope that you may find our response satisfactory and raise your score accordingly.
>
> **Q1-a: a big fuzz about using the RMD, even though simple K-means clustering already helps** It is true that K-means clustering can already reveal the benefit of using our proposal. As the reviewer already pointed out, our main idea is not limited to one specific sample difficulty measure. The fact that different reasonable measures provided gains is a positive evidence to our idea. Nevertheless, we believe it is still of technical interest to present RMD (see our next answer). We will definitely consider reviewer's suggestion on improving the presentation of sample difficulty measuring method.
>
> **Q1-b: why is RMD a better measure of sample difficulty than K-Means and MD?** These three metrics are all based on the Gaussian assumption in the feature space. The key difference between RMD and K-means/MD lies in the relative distance calculation. Both K-means/MD scores the sample difficulty based on the distance towards the mean mode, whereas RMD examines the relative distances towards two properly chosen mean modes, i.e., class-agnostic and class-specific mean. The relative distance reflects not only if the feature is typical (close to the class-specific mean), but also if it is discriminative (far from the class-agnostic mean). Both K-means and MD mostly concern the first aspect, thus perform worse than RMD, esp. when taking care of fine-grained classification benchmarks such as ImageNet1k. In ImageNet1k, many classes could share some common features, e.g., there are many sub-classes of elephants in ImageNet1k and they all look alike in some aspects. To confidently classify one elephant as Tusker, it must have typical features that are also discriminative enough from the rest of the classes. The metrical difference using RMD can filter out the influence from shared typical features, i.e., features close to the class-agnostic mean mode. We will make it clearer in the next version. We also add an illustrative example in Figure I of the attached rebuttal PDF in our global response, which hopefully visualizes well the above description.
>
> **Q2: Why do you take the maximum over i in Eq. 10?** By taking the maximum over "i", we make the score $s(x,y)$ upper-bounded by one. Overall, the score is constrained to the value range between 0 and 1, which is normalized and avoids potential numerical issues.
>
> **Q3: Table 4: misses a description of what the columns represent:** Thanks for pointing this out. Each column is associated with a value for the hyper-parameter $\alpha$. We will add this information upon revision.
>
> **Q4: Error bars (variance) of experimental results:** Thanks for the suggestion. We will add ``std'' of experimental results upon revision. At this point, we first report std of ACC and ECE for CE, ER and our method in Table Ⅰ of the attached rebuttal PDF in our global response. We can observe that the proposed method is still performing the best within the standard deviation. Our std is generally comparable to other methods.
>
> **Q5: What is the runtime of calculating RMD?** We can calculate RMD of each sample once before training to save computing overhead (as stated in line 213), so training and inference overhead is basically the same as other methods. As for the running time of calculating RMD of all training samples before training, we report results in Table A. It should be noted that we only need to calculate RMD once for a specific downstream task.
>
> Tab. A: The running time of calculating RMD of the entire training dataset.
> | Dataset          | CIFAR-10/100 | ImageNet-1k |
> |------------------|--------------|-------------|
> | Running time (s) | 94s          | 2068s       |
>
> **Q6: How well does the Gaussian assumption fit?** We chose the Gaussian assumption based on the following considerations. First, it leads to a simple distance measure for quantifying the sample difficulty, i.e., examining mean and variance. Second, it has several empirical supports from other tasks. For instance, to assess the image synthesis quality, the standard metric: FID essentially measures the difference between the real and synthetic data distributions based on multivariate Gaussian modeling in the feature space of Inception v3 or CLIP. Another example is the use of Gaussian assumption for deriving the OOD detection metric, e.g., in [*1].
>
> We would further note that our method is not limited by Gaussian assumption. We contributed the idea of modeling the feature distribution for sample difficulty quantification. Gaussian assumption is an example, which led to convincing gains. Nevertheless, one may resort to more sophisticated modellings for further improvements.
>
> Reference:
>
> [*1] Contrastive Training for Improved Out-of-Distribution Detection, Jim et al., arXiv:2007.05566

---

### Official Review · Reviewer_Ye8r · 2023-07-06

**Soundness:** 2 fair
**Presentation:** 3 good
**Contribution:** 2 fair
**Rating:** 5
**Confidence:** 3

**Summary:**

This paper proposes a difficulty-aware uncertainty regularization approach, which first pre-defines the difficulty of each training sample and then differently regularizes training samples during training. To quantify the sample difficulty, the authors utilize pre-trained large models like CLIP to extract feature vectors, and the relative Mahalanobis distance is computed on a collection of these feature vectors.

**Strengths:**

1. The paper is well-written and exhibits clarity in its presentation. The visual results help to comprehend the concept of sample difficulty.
2. The effectiveness of considering sample difficulty in classification tasks has been empirically verified against other uncertainty regularization methods. In particular, the ablation study conducted on the regularizing strength hyperparameter $\alpha$ in Section 5.4 clearly demonstrates that the proposed sample-specific regularization approach outperforms ER, which performs global regularization.

**Weaknesses:**

1. The current version of the paper solely presents the average value obtained from five trials without including information about the standard deviation. It is highly recommended to include error bars.
2. The proposed methodology heavily depends on CLIP as a key component. Specifically, when using ViT-B and MAE-ViT-B, the accuracies achieved in Table 5 are 73.59% and 73.81%, respectively, which are lower than the accuracy of 74.02% achieved by the Poly baseline presented in Table 1. At this stage, it is challenging to assert that the proposed method can be applied universally to any large-scale pre-trained models.

**Questions:**

1. The overall scheme that a pre-trained model provides supervision for a given data reminds the knowledge distillation. However, the authors stated "knowledge distillation will not solve the problem" in lines 29-30. Could you make an additional statement on this?
2. The reason for the absence of the CRL as a baseline in the main tables, specifically Tables 2 and 3, is unclear or not readily apparent.
3. Is there a specific justification for not including C10 in the misclassification detection experiment?

Miscellaneous:
1. Section 5.4. "roubustness" should be "robustness."
2. Table 4. "under different on ImageNet1k" should be "under different $\alpha$ on ImageNet1k."
3. It would be nice to update the figures in the paper to utilize colors that are accessible for individuals with color blindness. Specifically, figures 4, 7, and 11 might pose challenges for individuals with red-green color blindness.

**Limitations:**

The authors did not address the limitations.

---

> ### Author Rebuttal · Authors · 2023-08-09
>
> We thank you for finding our work clearly motivated and clarified, as well as providing valuable suggestions to further improve our paper. We address the detailed concerns below, by including the suggested standard deviation, a new experiment to further demonstrate the applicability beyond CLIP, and clarification on various points. We hope that you may find our response satisfactory and raise your score accordingly.
>
> **Q1: No standard deviation (error bars) about experimental results:**  We agree this is important. Our reported numbers are the mean values based on 5 runs for CIFARs and 3 runs for ImageNet1k. We will also add ``std'' in the final version. At this point, we first report std of ACC and ECE for CE, ER and our method in Table Ⅰ of the attached rebuttal PDF in our global response. We can observe that the proposed method is still performing the best within the standard deviation, and our std is generally comparable to other methods.
>
> **Q2: It is challenging to assert that the proposed method can be applied universally to any large-scale pre-trained models:** We disagree with our highest respect. In fact, while VIT-B and MAE-VIT-B in Tab. 5 are not as good as CLIP, they both provide convincing improvements in ECE. Then, among methods in Tab. 1 that achieve similar ECEs as them, they are better in terms of ACC. It is true that they both underperform PolyLoss in ACC. However, PolyLoss severely compromises ECE for the gain, whereas our method combined with different pre-trained models does not suffer from such a tradeoff, and improves both metrics over the baseline "CE". Furthermore, we perform a new experiment, using recent DINOv2 [*1] as a pre-trained model to quantify sample difficulty, and report ACC and ECE on CIFAR-10/100 in Table Ⅱ of the attached rebuttal PDF in our global response. While both CLIP and DINOv2 were trained on large-scale datasets, they took different training procedures. As shown, our method based on DINOv2 outperforms all baselines in Tab. 1 (including PolyLoss). The achieved gains are similar to that of using CLIP. Hence, our method itself can work with other pre-trained models.
>
> Reference:
>
> [*1] DINOv2: Learning Robust Visual Features without Supervision, Oquab et al., arXiv:2304.07193
>
> **Q3: More elaboration on the statement on "knowledge distillation will not solve the problem" in lines 29-30:** Sample difficulty measure can be regarded as a type of data annotation complementary to the ground-truth label. Solely relying on the ground-truth label, the cross entropy loss essentially induces the model to overfit the 0/1 loss, i.e., matching the label with 100% confidence. This is also the case when people use the cross entropy loss to train the teacher model as well as the student model with the feature matching with the trained teacher model. Therefore, this overfitting issue does not simply go away with knowledge distillation. Our method tries to address the issue by adding additional annotation to each sample, and modifying the training loss with instance-adaptive entropy regularization.
>
> Furthermore, we do not intend to make sure the "student" (i.e., the actual classifier) mimics the "teacher", neither in the feature space nor the logit space. Behaving like the teacher does not necessarily lead to better "uncertainty", where knowledge distillation solutions primarily focus on accuracy. It is an interesting venue that requires dedicated investigation, which however is beyond the scope of our work. We will improve our wording upon revision.
>
> **Q4: The reason for the absence of the CRL as a baseline in the main tables, specifically Tables 2 and 3:** We took CLR as an approach that exploits sample difficulty measures for uncertainty in ablation studies, thus we primarily compared them among methods based on sample difficulty, i.e., Sec. 5.5 for comparison of different sample difficulty measures. Of course, we are happy to provide results about misclassification and OOD detection for CRL. As shown in Tab. A and Tab. B below, our method still outperforms CRL regarding misclassification and OOD detection. We will add detailed results in the final version.
>
> Tab.A: The comparison of misclassification detection performance (%) for CRL and our method.
> | Dataset    | Method   | FPR-95\%$\downarrow$ | AUROC $\uparrow$     | AUPR $\uparrow$      |
> |------------|----------|----------------------|----------------------|----------------------|
> |            |          |       | MSP / Entropy  |
> | C100  | CRL      | 44.80/46.08   | 86.67/85.98    | 95.55/95.49 |
> |            | Proposed | **42.71/43.22** | **87.50/87.03** | **96.10/96.02** |
> | ImageNet1k | CRL | 46.03/48.01   | 86.11/84.33  | 94.41/93.89  |
> |            | Proposed | **45.69/46.72** | **86.53/85.23** | **94.76/94.31** |
>
> Tab.B: The comparison of near OOD detection performance (%) for CRL and our method.
> |$D_{in}$ /$D_{out}$    | Method   | FPR-95\%$\downarrow$ | AUROC  $\uparrow$ | AUPR  $\uparrow$  |
> |------------|----------|----------------------|----------------------|----------------------|
> |   |    |     | MaxLogit / Entropy   |
> | C100/C10   | CRL      | 58.13/58.54   | 79.91/80.13  | 81.75/81.89  |
> |            | Proposed | **55.48/55.60** | **80.20/80.72** | **82.51/82.84** |
> | ImageNet1k /iNaturalist | CRL | 35.07/34.65  | 90.11/90.32  | 97.96/97.81  |
> |            | Proposed | **32.17/34.19** | **91.03/90.65** | **98.03/97.99** |
>
> **Q5: Is there a specific justification for not including C10 in the misclassification detection experiment?** Misclassification detection is more relevant for scenarios with low classification accuracy, whereas the accuracy of C10 has already reached about 95\%. Therefore, we focused on C100 and ImageNet1k, which are more interesting/challenging benchmarks for misclassification detection.
>
> **Q6: Editing errors and colors in figures 4, 7, and 11:** Thank you for your suggestions, and we will update them in the final version.

---

> > ### Comment · Reviewer_Ye8r · 2023-08-12
> >
> > Thank you for the authors' efforts. Incorporating supplementary "difficulty" characteristics into an existing dataset via pre-trained models represents a straightforward yet intriguing strategy to offer extra insights into the data. The supplementary experiments involving DINOv2 work to reaffirm this notion, and integrating them into the main text would be beneficial. More precisely, showcasing the primary results through ViT (supervised), MAE-ViT, CLIP-ViT, and DINO-ViT would make the paper solid and underscores its alignment with the title (i.e., pre-trained models). I am pleased to raise my score and anticipate addressing the concerns raised during the rebuttal phase in the final manuscript.

---

> > > ### Author Response · Authors · 2023-08-13
> > > **Thank you for raising the score!**
> > >
> > > We are pleased to know that Reviewer Ye8r finds our rebuttal satisfactory and raises the score. We will address the concerns in the final manuscript, as stated in the rebuttal.

---

### Official Review · Reviewer_FmxN · 2023-07-07

**Soundness:** 3 good
**Presentation:** 3 good
**Contribution:** 2 fair
**Rating:** 6
**Confidence:** 4

**Summary:**

In settings where deep neural networks are used for critical tasks, it is crucial to ensure that they are calibrated and capable of reliable predictions. It is desirable to have the ability to measure the confidence of the model's predictions and reject those that have high uncertainty. To achieve this, the authors suggest using neural networks that have been pre-trained on large and diverse datasets as calibration auxiliaries for downstream tasks.

Through extensive experimentation on benchmarking datasets such as ImageNet, the authors demonstrate that using pre-trained models such as CLIP to define the relative Mahalanobis Distance in feature space is a valid measure of sample difficulty. Additionally, they introduce a novel regularization term that accounts for the difficulty of individual samples. In section 5, they provide an extensive evaluation of their proposed learning objective in comparison to other standard regularizers. While difficulty-aware regularization can be somewhat costly, it ultimately reduces the expected calibration error.

**Strengths:**

Some strengths of the work:
**Originality**: Researchers are actively studying how to measure the difficulty of individual instances in machine learning to improve its reliability. The authors have introduced and proven the effectiveness of RMD as a reliable method for measuring this difficulty.

**Clarity**: I found the paper well-written and straightforward to understand. The authors have provided ample empirical evidence to support their idea of sample difficulty, and they have included a comprehensive discussion while comparing it to relevant baselines.

**Quality**: Understanding the limitations of existing algorithms on complex problems requires measuring sample difficulty. This research introduces innovative regularizers that enhance the models' calibration in downstream training.

**Significance**: Alongside establishing robust benchmarks and regularizers for calibrated training, the authors also find that self-supervised learning algorithms (such as MAE) learn richer representations that better estimate hardness.

**Weaknesses:**

1. When it comes to measuring sample difficulty, the effectiveness of using large-scale pretrained datasets depends on the degree of domain/distribution shift in the downstream task. For example, it remains uncertain whether the same measures can be applied to medical imaging tasks as compared to natural image classification.

**Questions:**

1. The authors explore example difficulty in the context of classification tasks. Can similar measures be extended to dense-prediction tasks?

**Limitations:**

It's important to mention that measuring the difficulty of samples adds extra computing workload. To provide clarity to readers, the authors are encouraged to discuss this topic and any potential limitations of their proposed work.

---

> ### Author Rebuttal · Authors · 2023-08-09
>
> We thank you for finding our work innovative, well-written and straightforward to understand, and showing convincing validation, as well as providing valuable comments. We answer the specific questions below. We hope you will find our response satisfactory and raise your score accordingly.
>
> **Q1: It remains uncertain whether the same measures can be applied to medical imaging tasks as compared to natural image classification:** We understand the concern may arise from the fact that CLIP may be inadequate for the medical domain, as medical images can be OOD (out-of-distribution) to CLIP. Nevertheless, we would like to note that our procedure of deriving sample difficulty is compatible with other pre-trained models. For instance, we add a new experiment to demonstrate that our method is plug-and-play to switch from CLIP to DINO v2 [*1], delivering comparable gains as on the natural image classification benchmarks (See Table II in the attached rebuttal PDF in our global response). For the medical domain, MedCLIP [*2] can be a more interesting alternative than CLIP/DINO v2 for practicing our method. We will include this discussion in the final version.
>
> Reference:
>
> [*1] DINOv2: Learning Robust Visual Features without Supervision, Oquab et al., arXiv:2304.07193
>
> [*2] MedCLIP: Contrastive Learning from Unpaired Medical Images and Text, Wang et al., EMNLP 2022.
>
>
> **Q2: Can sample difficulty measures be extended to dense-prediction tasks?** It is an interesting perspective. The challenge for dense-prediction tasks lies in multiple objects' coexistence in one image. To use our measure, getting the feature per object instance is important. Taking object detection as an example, a natural way to use our method would be to use the ground-truth bounding boxes of the training samples to extract the features per object instance (e.g., using ROI align) before scoring the sample difficulty of the training set. The sample difficulty can then be used for regularizing the classification head, which is also on each object instance level. We will include such discussion in the final version.
>
> **Q3: Mention the computation overhead and discuss potential limitations of their proposed work:** We will add the computation overhead into the implementation details. Briefly, we can calculate per-sample difficulty score once before training to save computing overhead (as stated in line 213), so training and inference overhead is basically the same as other methods. Overall, we find the computation overhead very affordable, not being a limiting factor of our method. As for a general limitation discussion, we have included some in the conclusion in the form of future extensions of our method. Upon revision, we will incorporate the discussion in Q1 as well.

---

### Author Rebuttal · Authors · 2023-08-09

Firstly, we would like to express our gratitude for the thoughtful reviews, which help to further improve our paper. We are pleased that the reviewers found our paper to be **novel (innovative)** (Reviewers FmxN, YeMZ, Lxfs), **well-written and convincing** (All), **straightforward to understand** (Reviewer FmxN), the studied problem is **important** and the training process is **flexible** (Reviewer Lxfs), and our experiments to be **strong and convincing** (Reviewer 93FC).

Secondly, for some common concerns, 1) **no standard deviation (error bars) about experimental results**, we provide Table Ⅰ in the uploaded rebuttal PDF, which shows the std is on a par with other methods; 2) **Why is RMD a better measure of sample difficulty than K-Means and MD**, we provide detailed explanations to reviewers individually. In addition, we also add an illustrative example in Figure A of the uploaded rebuttal PDF, which hopefully visualizes well the benefit of using relative distance; 3) **Whether our method can be applied to other large-scale pre-training models**, we perform a new experiment, using recent DINOv2 as a pre-trained model to quantify sample difficulty, and report ACC and ECE on CIFAR-10/100 in Table Ⅱ of the uploaded rebuttal PDF. While both CLIP and DINOv2 were trained on large-scale datasets, They took different training procedures. As shown, our method based on DINOv2 also improves over the baselines in both ACC and ECE, delivering superior performance similar to CLIP. Hence, our method itself is not CLIP-specific, and can work with other pre-trained models.

We hope this message provides a good summary of the reviews and our responses. We further address the comments of each reviewer individually.

---

### Author Response · Authors · 2023-08-19
**Looking forward to further feedback**

Dear Reviewers,

Thank you again for your thorough reviews. We are glad that both Reviewer 93FC and Ye8r found our rebuttal satisfactory and increased the score accordingly. We invite the remaining three reviewers (FmxN, YeMZ, Lxfs) to check our rebuttal. Please kindly note that the discussion phase is close to the end (Aug 21st).  If you have any further concerns or requests, we would be very happy to address them in the author-reviewer discussion period. If all your concerns have been resolved, it is much appreciated if you may raise the rating of our work.

Best,

The Authors

---

### Decision · Program_Chairs · 2023-09-21

**Decision:**

Accept (poster)

**Comment:**

After extensive discussion with the authors, the reviewers all agree that this is a solid contribution that should be accepted. I look forward to seeing the final version with the reviewers' suggestions incorporated.